TECHNICAL RELEASE

# Julearn: an easy-to-use library for leakage-free evaluation and inspection of ML models

Sami Hamdan[1,2], Shammi More[1,2], Leonard Sasse[1,2,3], Vera Komeyer[1,2], Kaustubh R. Patil[1,2], Federico Raimondo[1,2,*] and for the Alzheimer's Disease Neuroimaging Initiative[†]

1 Institute of Neuroscience and Medicine (INM-7: Brain and Behaviour), Research Centre Jülich, Germany
2 Institute of Systems Neuroscience, Heinrich Heine University Düsseldorf, Germany
3 Max Planck School of Cognition, Stephanstrasse 1a, Leipzig, Germany

**Submitted:** 13 November 2023

\* Corresponding author. E-mail:
f.raimondo@fz-juelich.de

† Data used in preparation of this article were obtained from the Alzheimer's Disease Neuroimaging Initiative (ADNI) database (adni.loni.usc.edu). As such, the investigators within the ADNI contributed to the design and implementation of ADNI and/or provided data but did not participate in analysis or writing of this report. A complete listing of ADNI investigators can be found at: http://adni.loni.usc.edu/wp-content/uploads/how_to_apply/ADNI_Acknowledgement_List.pdf

Preprint submitted at https://doi.org/10.48550/arXiv.2310.12568

## ABSTRACT

The fast-paced development of machine learning (ML) and its increasing adoption in research challenge researchers without extensive training in ML. In neuroscience, ML can help understand brain-behavior relationships, diagnose diseases and develop biomarkers using data from sources like magnetic resonance imaging and electroencephalography. Primarily, ML builds models to make accurate predictions on unseen data. Researchers evaluate models' performance and generalizability using techniques such as cross-validation (CV). However, choosing a CV scheme and evaluating an ML pipeline is challenging and, if done improperly, can lead to overestimated results and incorrect interpretations. Here, we created julearn, an open-source Python library allowing researchers to design and evaluate complex ML pipelines without encountering common pitfalls. We present the rationale behind julearn's design, its core features, and showcase three examples of previously-published research projects. Julearn simplifies the access to ML providing an easy-to-use environment. With its design, unique features, simple interface, and practical documentation, it poses as a useful Python-based library for research projects.

**Subjects** Software and Workflows, Neuroscience, Machine Learning

## INTRODUCTION

Machine Learning (ML) is fast becoming an indispensable tool in many research fields. It is rapidly gaining increasing importance within neuroscience, where it is used for understanding brain-behavior relationships [1], predicting disease status and biomarker development using diverse data modalities such as Magnetic Resonance Imaging (MRI) and electroencephalogram. Such ML applications are driven by the availability of big data and technological advances. However, for domain experts, acquiring relevant ML and programming skills remains a significant challenge. This underscores the need for user-friendly software solutions accessible to domain experts without extensive ML training. Such solutions would enable them to quickly evaluate ML approaches.

An ML application aims to create a model that provides accurate predictions on new unseen data—i.e., a generalizable model. In this context, the goal of a research project is usually to demonstrate that a generalizable model exists for the prediction task at hand. As a single set of samples is usually available, this goal is achieved by assessing the



generalization performance by training the model on a subset of the data and testing it on the hold-out test data. If the model performs well on the test data, then the researcher concludes that the prediction task can be solved in a generalizable manner. One of the most prominent approaches to estimating the generalization performance is cross-validation (CV). CV is a systematic subsampling approach, which trains and tests ML pipelines multiple times using independent data splits [2]. The average performance over the splits is taken as an estimate of generalization. To achieve good performance or other aims, like data interpretation, it is often necessary to perform additional data processing, for example, feature selection. This results in an ML pipeline that performs all the needed operations from data manipulations, training and evaluation. Choosing a CV scheme and evaluating an ML pipeline can be challenging, and if improperly used, it can lead to incorrect results and misguided insights. This underscores the need for user-friendly software solutions accessible to domain experts without in-depth ML and programming training. Problematically, a common outcome of pitfalls is an overestimation of the generalization performance when using CV, i.e., models are reported as being more accurate than what they actually are. Here, we highlight two common pitfalls: data leakage and overfitting of hyperparameters.

Data leakage occurs when the separation between the training and test data is not strictly followed. For instance, using all available data in parts of an ML pipeline breaks the required separation between training and test data. Such data leakage invalidates the complete CV procedure, as information on the testing set is available during training. For example, one might apply a preprocessing step, like $z$-standardization or Principal Component Analysis (PCA), on the complete dataset before splitting the data. As the preprocessing step is informed about the test data, the later created and transformed training data will reflect the test data. Therefore, the learning algorithm can leverage this leaked test set information through preprocessing and memorization instead of building a predictive model, thus inflating the generalization estimation of CV. Most problematically, data leakage can happen in many ways through programming errors or lack of awareness of this danger.

A similar pitfall can occur when tuning hyperparameters by first observing their test set performance. Hyperparameters are parameters not learnable by the algorithms, which greatly impact their prediction performance. To tackle this optimization problem, many practitioners repeat a simple CV to evaluate the test set performance of different hyperparameter combinations. Problematically, both tuning and estimating out-of-sample performance on the same test data breaks the clear distinction between training and testing, as one both optimizes and evaluates the ML pipeline on the same test set. Notably, this can happen very quickly over the natural progression of research projects while iterating through ideas of appropriate hyperparameters. The solution to this pitfall is to select the hyperparameters and evaluate the out-of-sample performance in different data splits, which can be achieved by using a nested CV. In conclusion, both pitfalls can happen easily and without malicious intent through a lack of ML or programming experience. We developed the open-source Python package julearn to allow field experts to circumvent these pitfalls by default while training and evaluating ML pipelines.

While ML experts can navigate these and other pitfalls using expert software, such as scikit-learn (RRID:SCR_002577), domain experts might not always be aware of the pitfalls or how to handle them. This is why we created julearn, to provide an out-of-the-box solution,



preventing common mistakes, usable by domain experts. Julearn was created to be easy to use, accessible for researchers with diverse backgrounds, and to create reproducible results. Furthermore, we engineered julearn so it is easy to extend and maintain, in order to keep up with constantly evolving fields such as neuroscience and medicine. The accessibility and usability aspects of julearn were decided to be at the core, as we aimed to help researchers apply ML. We accomplished this through a careful design of the Application Programming Interface (API), comprising only a few simple key functions and classes to create and evaluate complex ML pipelines. Furthermore, we added several utilities that allow investigators to gain a detailed understanding of the resulting pipelines. In order to keep julearn up to date, we built it on top of scikit-learn [3, 4] and followed common best practices of software engineering, like unit testing and continuous integration.

## METHODS

### Basic usage

Julearn is built on top of scikit-learn [3, 4], one of the most influential ML libraries in the Python programming language. While scikit-learn provides a powerful interface for programmers to create complex and individualized ML pipelines, julearn mainly adds an abstraction layer, providing a simple interface for novice programmers. That is, a user-friendly and easy-to-program API, tailored for users with basic programming skills or limited knowledge of ML who wish to start with ML or evaluate complex ML pipelines in an error-free way. Note that while scikit-learn is a general ML Library, julearn focuses on so-called supervised ML tasks, which include any prediction task with known labels while training and evaluating pipelines. Therefore, pipelines in the context of julearn always refer to supervised ML pipelines. Importantly, rather than posing itself as a replacement or competitor, julearn aims to enhance scikit-learn's features while providing access to scikit-learn's functionality for supervised ML. Consequently, it is also possible to use a custom scikit-learn compatible model.

To achieve a simple interface for supervised ML problems, we implemented a core function called `run_cross_validation` to estimate a model's performance using CV. In this function, the user specifies the data, features, target, preprocessing and model name to evaluate as an ML pipeline in a leakage-free, cross-validated manner. We chose the popular and simple tabular data structure of Pandas' `DataFrame` [5] for both the input data and the output of `run_cross_validation`. This makes preparing the input, as well as inspecting and analyzing the output of julearn, simple and transparent.

Furthermore, our API provides arguments for feature and target name(s) referring to the columns of the input data frame. To use any of julearn's ML algorithms, one only needs to provide their name to the `model` argument of `run_cross_validation`. Here, julearn will select the model according to the provided `problem_type` of either classification or regression. Similarly, one can provide any of the supported preprocessing steps to `run_cross_validation` by name. These steps are executed in a CV-consistent way without the risk of data leakage. Such an interface simplifies the construction and use of ML pipelines, in contrast to scikit-learn, where one must import different ML models depending on the problem type, create a pipeline using both the imported preprocessing steps and the ML model and finally use the `cross_validate` function (Figure 1).

While julearn does not aim to replace scikit-learn, it tries to simplify specific use cases, including the creation of more complex supervised ML pipelines that need hyperparameter

A)

```python
from julearn import run_cross_validation
run_cross_validation(
    X=X,   # Feature names
    y=y,   # Target variable name
    data=data, # Pandas DataFrame to use
    preprocess=["zscore"],   # Preprocessing steps
    model="svm",   # Learning algorithm
    problem_type="classification",
    X_types={"continuous": X}  # X_types optional here
)
```

B)

```python
from sklearn.model_selection import cross_validate
from sklearn.svm import SVC   # SVR in case of regression
from sklearn.preprocessing import StandardScaler
from sklearn.pipeline import make_pipeline

# Create a scikit-learn pipeline with the required steps
pipeline = make_pipeline(StandardScaler(), SVC())

# Run cross_validate with the training features and targets
cross_validate(X=data.loc[:, X], y=data.loc[:, y], estimator=pipeline)
```

**Figure 1.** Implementation of a simple CV pipeline using julearn (A) in contrast to scikit-learn (B). The julearn pipeline needs only one import, while scikit-learn needs multiple ones. Furthermore, scikit-learn needs to import the Support Vector Machine differently depending on the problem type, while julearn chooses the correct one based on the problem type. The differences between julearn and scikit-learn are most influential for inexperienced programmers who aim to create (complex) supervised ML pipelines. Julearn builds upon scikit-learn by providing a simple interface that does not need any awareness of how to compose and find different classes.

tuning or preprocessing a subsample of features. This means that julearn can automatically use nested CV for proper performance assessment in the context of hyperparameter tuning [6] and apply preprocessing based on different feature types. These feature types include distinctions like categorical vs continuous features or grouping variables, which can even be used to do confound removal on a subsample of the data.

## Model comparison

In ML applications, there is no standard or consensus of what a good or acceptable performance is, as this usually depends on the task and domain. Thus, the process of developing predictive models involves comparing models, either to null or dummy models, or to previously published models (i.e., benchmarking). Given that CV produces estimates of the model's performance and that, depending on the CV strategy, these estimates might not be independent from each other, special methods are required to test and conclude if the performance of two models is different or not. For this reason, julearn `run_cross_validation` output has additional information that can be used to make more accurate model comparisons. Furthermore, it provides a stats module, which implements a student's *t*-test corrected for using the same CV approach to compare multiple ML pipelines [7]. This correction is necessary as CV leads to a dependency between the folds,

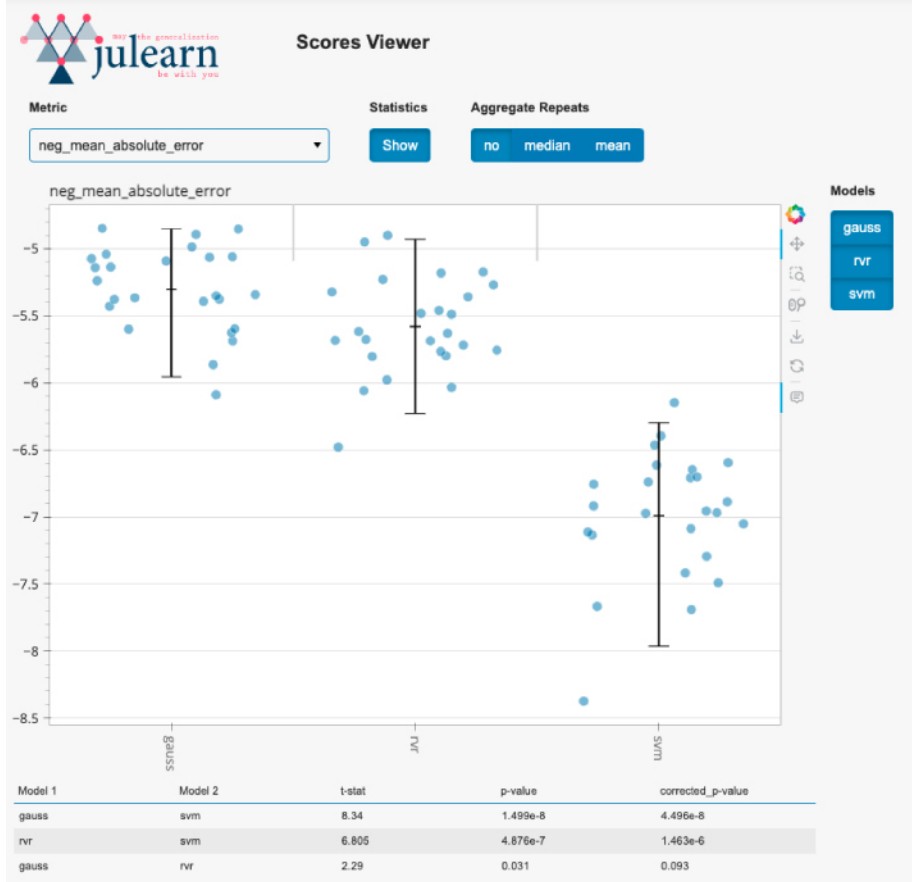

**Figure 2.** Screenshot of the julearn scores viewer, depicting the negative mean absolute error in age prediction from gray matter volume. Each dot represents the negative mean absolute error of each CV fold (5 times, 5-folds). Each column represents a different model: Gaussian Process Regression (GPR) (gauss), Relevance Vector Regression (RVR) (rvr) and Support Vector Regression (SVR) (svm). Black lines indicate the mean and 95% confidence intervals. The table at the bottom shows the pairwise statistics using the corrected *t*-test.

i.e., each iteration's training set overlaps with the other ones. To gain a detailed view of the models' benchmark, one can also use julearn's inbuilt visualization tool (see Figure 2 for example).

## Feature types

One key functionality that julearn provides, which is currently lacking in ML libraries such as scikit-learn, is the ability to define feature types. This allows researchers to define sets of variables and do selective processing, needed when dealing with categorical or confounding variables. For this matter, julearn introduces the `PipelineCreator` to create complex pipelines in which certain processing steps can be applied to one or more subsets of features. Once the pipeline is defined, users need to provide a dictionary of any user-defined type and the associated column names in their data as the `X_types` argument. Such functionality allows to implement complex pipelines that transform features based on their *type*, e.g., standardizing only continuous features and then deconfounding both continuous and categorical features.

### Hyperparameter tuning

As mentioned previously, hyperparameter tuning should be performed in a nested CV to not overfit the predictions of a given pipeline. The `PipelineCreator` can be used to specify sets of hyperparameters to be tested at each individual step by just using the `add` method (Figure 3). Being able to first define a pipeline and its hyperparameters with the `PipelineCreator`, and to then train and evaluate this pipeline with `run_cross_validation`, makes performing leakage-free nested CV easy. In this nested CV, all hyperparameters are optimized in an inner CV using a grid search by default. This default, like most of julearn's defaults, can be easily adjusted by providing any compatible searcher in the `run_cross_validation`'s `model_params` argument. This is a drastic simplification compared to a typical scikit-learn workflow, where one must create the pipeline manually by combining different objects, wrap it inside a `GridSearchCV` object, and define the hyperparameter options separately from the pipeline itself, using a complex syntax. Lastly, scikit-learn's `GridSearchCV` object must be provided to its `cross_validate` function.

### Inspection and analysis

Inspection of ML pipelines is crucial when working in fields such as neuroscience and medicine, as concepts like trustworthy ML are heavily dependent on the ability to draw insights and conclusions from models. For this purpose, one needs to be able to inspect and verify each pipeline step, check parameters, and evaluate feature importances and further properties of ML pipelines. Julearn includes two functionalities: a `preprocess` function and an `Inspector` class. The `preprocess` function allows users to process the data up to any step of the pipeline, allowing them to check how the different transformations are applied. For example, a user might be interested in examining the PCA components created or the distribution of features after confound removal (see Figure 4 for example). The `Inspector` object, on the other hand, allows us to inspect the models after estimating their performance using CV. It helps users to check fold-wise predictions and obtain both the hyper- and fitted parameters of the trained models (see Figure 5 for example). This enables users to verify the robustness of the different parameter combinations and evaluate the variability of the performance across folds. Ongoing efforts to increase julearn's inspection tools encompass integrating tools for explainable Artificial Intelligence (AI), such as SHAP [8].

### Neuroscience-specific features

In addition to julearn's field-agnostic features, we also provide neuroscience-specific functionalities. Confound removal in the form of confound regression, which is popularly used in neuroscience, was implemented as the `ConfoundRemover`. This confound regression can be trained on all features or only on specific subsamples defined by a grouping variable, i.e., allowing neuroscientists to only train it on healthy participants as proposed in Dukart *et al.* [9]. Additionally, we have included the Connectome Based Predictive Modelling (CBPM) algorithm [10]. This transformer aggregates features significantly correlated with the target into one or two features. This can be done separately for the positively and negatively correlated features. Aggregation can be done using any user-specified aggregation function, such as summation or mean. We plan to add more neuroscience specific features, such as the integration of harmonization techniques, currently developed in a separate project (*juharmonize*).



A)

```python
from julearn import run_cross_validation, PipelineCreator

# Create a pipeline with several hyperparameters per step
creator = PipelineCreator(problem_type="classification")
creator.add("zscore", with_mean=[True, False])
creator.add("pca", n_components=2)
creator.add("svm", C=[1,2], degree=[3,4])

run_cross_validation(
    X=X, y=y, data=data,  # Specify features, target and data
    model=creator,  # Use the creator as model
    X_types={"continuous": X},  # X_types optional
)
```

B)

```python
from sklearn.model_selection import cross_validate, GridSearchCV
from sklearn.svm import SVC  # SVR in case of regression
from sklearn.preprocessing import StandardScaler
from sklearn.decomposition import PCA
from sklearn.pipeline import make_pipeline

# Create a scikit-learn pipeline with the required steps
pipeline = make_pipeline(StandardScaler(), PCA(), SVC())

# Define the parameter grid
param_grid = {
    "standardscaler__with_mean": [True, False],
    "pca__n_components": [2],
    "svc__C": [1,2],
    "svc__degree": [3,4]
}

# Create a GridSearchCV estimator
grid_pipeline = GridSearchCV(estimator=pipeline, param_grid=param_grid)

# Run cross_validate on the GridSearch
cross_validate(X=data.loc[:, X], y=data.loc[:, y], estimator=grid_pipeline)
```

**Figure 3.** Example of julearn (A) and scikit-learn (B) training a typical ML pipeline in a CV consistent way. Both use a grid search to find optimal hyperparameters. Note that julearn is able to specify the hyperparameters at the same time as it defines each step. On the other hand, scikit-learn needs all hyperparameters to be defined separately with a prefix indicating the step they belong to. This can become complex, especially when pipelines are nested and multiple prefixes are needed.

## Customization and extensibility

Julearn provides a simple interface to several important ML approaches but is also easily customizable. Each component of julearn is built to be scikit-learn compatible, meaning that any scikit-learn compatible model and transformer can be provided to `run_cross_validation` and `PipelineCreator`. Other `run_cross_validation` arguments, like `cv` and hyperparameter searchers, were implemented in a way to be extensible by any typical scikit-learn object. This customizability of julearn helps users extend their usage of julearn



```python
# Import required functions
from julearn import run_cross_validation
from julearn.inspect import preprocess
from julearn.pipeline import PipelineCreator

# Define the pipeline, including z-scoring and PCA
pipeline_creator = PipelineCreator(problem_type="regression")
pipeline_creator.add("select_variance", apply_to="*", threshold=0.15)
pipeline_creator.add("zscore")
pipeline_creator.add("pca", n_components=2)
pipeline_creator.add("rf", n_estimators=200)

# Run cross validation, returning the final model
scores, model = run_cross_validation(
    X=X,
    y=y,
    X_types=X_types,
    data=df,
    model=pipeline_creator,
    scoring=["r2", "neg_mean_absolute_error"],
    return_estimator="final",
    seed=200,
)

# Check how the model transforms X until the zscore step
X_after_zscore = preprocess(model, X=X, data=df, until="zscore")

# Check how the model transforms X until the PCA step
X_after_pca = preprocess(model, X=X, data=df, until="pca")
```

**Figure 4.** Example of the utility of the `preprocess` function. Once the model has been trained and evaluated using `run_cross_validation`, the user can verify how the data is transformed in the pipeline until a certain step. The whole functioning code as well as plots depicting the data points can be seen in julearn's documentation (Examples → Inspection → Preprocessing with variance threshold, zscore and PCA).

and prepares them for the case that they want to transition to scikit-learn to build unique expert level ML pipelines.

## EXAMPLES

To illustrate the functionality and quality attributes of julearn, we depict three independent examples, showing how the analysis described in previously-published research projects can be implemented with julearn.

## Example 1: prediction of age using Gray Matter Volume (GMV) derived from T1-weighted MRI images

### *Dataset*

We used T1-weighted (T1w) MRI images from the publicly available Information eXtraction from Images (IXI) dataset [11] (IXI, N = 562, age range = 20–86 years) for age estimation similar to Franke *et al.* [12].



```python
# Import required functions
from julearn import run_cross_validation
from julearn.inspect import preprocess
from julearn.pipeline import PipelineCreator

# Define the pipeline
creator = PipelineCreator(problem_type="classification")
creator.add("zscore")
creator.add("svm")

# Run cross validation, returning the final model and inspector
scores, model, inspector = run_cross_validation(
    X=X,
    y=y,
    data=df_iris,
    model=creator,
    return_inspector=True,
    cv=cv,
)

# Obtain a dataframe with the fold-wise predictions
cv_predictions = inspector.folds.predict()

# Obtain the trained model from the first fold
fold0_model = inspector.folds[0].model
```

**Figure 5.** Example of the usage of the `inspector`. The `run_cross_validation` can return the inspector, allowing the user to check the fold-wise predictions as well as the model parameters from each fold. A working example can be found on julearn's documentation (Examples → Inspection → Inspecting the fold-wise predictions).

### Image preprocessing

T1w images were preprocessed using the Computational Anatomy Toolbox (RRID:SCR_019184) version 12.8 [13]. The initial affine registration of T1w images was done with higher than default accuracy (accstr = 0.8), to ensure accurate normalization and segmentation. After bias field correction and tissue class segmentation, accurate optimized Geodesic shooting [14] was used for normalization (regstr = 1). We used 1 mm Geodesic Shooting templates and generated 1 mm isotropic images as output. Next, the normalized Gray Matter (GM) segments were modulated for linear and non-linear transformations.

### Feature spaces and models

A whole-brain mask was used to select 238,955 GM voxels. Then, smoothing with a 4 mm FWHM Gaussian kernel and resampling using linear interpolation to 8 mm spatial resolution was applied resulting in 3,747 features. We tested three regression models, GPR, RVR and SVR, using this feature space to predict age.

### Prediction analysis

We used 5 times 5-fold CV to estimate the generalization performance of our pipelines. Hyperparameters were tuned in the inner 5-fold CV. Features with low variance were removed (threshold $< 1 \times 10^{-5}$). PCA was applied on the features to retain 100% variance. The GPR model gave lowest generalization error (mean Mean Absolute Error (MAE) $= -5.30$ years), followed by RVR (MAE $= -5.56$) and SVR (MAE $= -6.98$). Corrected *t*-test revealed a significant difference between GPR and SVM ($p = 3.18 \times 10^{-9}$), and between RVR and SVM ($p = 8.19 \times 10^{-9}$). There was no significant difference between RVR and GPR ($p = 0.075$). Results can be visualized with julearn's scores viewer as depicted in Figure 2.

## Example 2: confound removal
### Dataset

For this example, we retrieved data conceptually similar to Dukart *et al.* [9]. We used the Alzheimer's Disease Neuroimaging Initiative https://adni.loni.usc.edu/ database including 498 participants and 68 features. We used age as a confound and the current diagnosis as the target. To simplify the task, we only predicted whether a participant had some form of impairment (mild cognitive impairment or Alzheimer's disease) or not (control).

### Prediction analysis

We aimed to conceptually replicate Figure 1 from Dukart *et al.* [9]. The authors proposed to train confound regression on the healthy participants of a study and then transform all participants using this confound regression. As part of their efforts, they compared two pipelines using the same learning algorithm (i.e., SVM) [15]. One pipeline was trained to directly classify healthy vs unhealthy participants without controlling for age, while a second pipeline was configured to first control for age using their proposed method: train the confound regression only on healthy participants. They evaluated the bias of age in the predictions of these models by comparing the age distributions of healthy vs unhealthy participants for each model's misclassifications. This was done by computing, for each pipeline, whether there is a significant age difference between these two groups of participants. They found a significant difference when not controlling for age, but not when controlling for age. With further experiments, they conclude that their method leads to less age-related bias. In this example, we replicated the comparison between the two SVMs. First, we built both pipelines using julearn and then compared their misclassified predictions to find the same differences (Figure 6).

While the first pipeline (without confound removal) is straightforward to implement, the second variant requires a complicated preprocessing step in which the confound removal needs to be trained on a subsample of one specific column of the data. Thanks to julearn's support for feature types, the whole procedure can be easily implemented by indicating which feature type are to be considered confounds (e.g., age), which column has the subsampling data (e.g., current diagnosis) and which values should be considered (e.g., healthy). Note that the difference between all subjects in age is significant for our larger, but not their smaller, sample, which can be attributed to the increased power due to the large sample size.

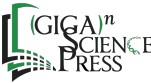

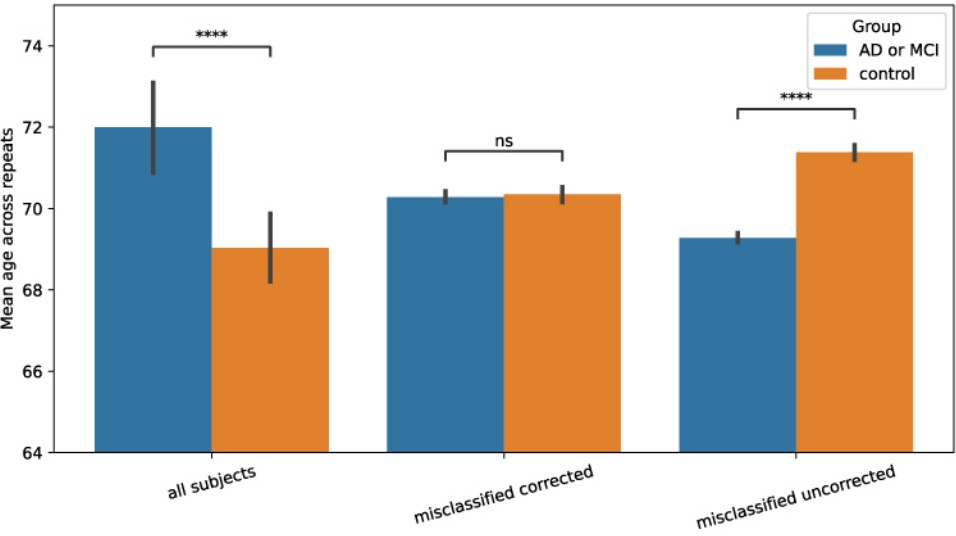

**Figure 6.** Replication of figure 1 in "Age characteristics of misclassified subjects using SVM" from Dukart *et al.* [9]. Performing a cross-validated confound removal trained only on the control group using julearn. Julearn greatly simplifies the process of training CV-consistent preprocessing steps based on characteristics like control vs experimental group. **** means a statistical significance at a *p*-value threshold of 0.0001 and ns that there is no statistical difference at that threshold.

## Example 3: prediction of fluid intelligence using connectome-based predictive modelling

### Dataset

We used data obtained from two resting-state functional Magnetic Resonance Imaging (rs-fMRI) sessions from the Human Connectome Project Young-Adult (HCP-YA) S1200 release [16]. The details regarding the collection of behavioral data, rs-fMRI acquisition, and image preprocessing have been described elsewhere [17, 18]. Here, we provide an overview. The scanning protocol for HCP-YA was approved by the local Institutional Review Board at Washington University in St. Louis. Retrospective analysis of these datasets was further approved by the local Ethics Committee at the Faculty of Medicine at Heinrich-Heine-University in Düsseldorf. We selected sessions for both phase encoding directions (left-to-right and right-to-left) obtained on the first day of HCP-YA data collection. Due to the HCP-YA's family structure, we selected 399 unrelated subjects (matched for the variable "Gender"), so that we could always maintain independence between folds during cross-validation. In line with Finn *et al.* [19], we filtered out subjects with high estimates of overall head motion (frame-to-frame head motion estimate (averaged across both day 1 rest runs; HCP-YA: MOVEMENT_RELATIVERMS_MEAN > 0.14). This resulted in a dataset consisting of 368 subjects (176 female, 192 male). Participants' ages ranged from 22 to 37 (mean = 28.7, standard deviation = 3.85). The two sessions of rs-fMRI lasted 15 min each, resulting in 30 min across both sessions. Scans were acquired using a 3T Siemens connectome-Skyra scanner with a gradient-echo EPI sequence (TE = 33.1 ms, TR = 720 ms, flip angle = 52°, 2.0 mm isotropic voxels, 72 slices, multiband factor of 8).

### Image preprocessing

Data from the rs-fMRI sessions in the HCP-YA had already undergone the HCP's minimal preprocessing pipeline [17], including motion correction and registration to standard space. Additionally, the Independent Component Analysis and FMRIB's ICA-based X-noiseifier (ICA-FIX) procedure [20] were applied to remove structured artefacts. Lastly, the 6 rigid-body parameters, their temporal derivatives and the squares of the 12 previous terms were regressed out, resulting in 24 parameters. In addition, we regressed out mean time courses of white matter, cerebro-spinal fluid and global signal, as well as their squared terms, the temporal derivatives of the mean signals and their squared terms as confounds, resulting in 12 parameters (4 for each noise component). The signal was linearly detrended and bandpass filtered at 0.01–0.08 Hz using `nilearn.image.clean_img`, The resulting voxel-wise time series were then aggregated using the Shen parcellation [19] consisting of 268 parcels. Functional Connectivity (FC) was estimated for each rs-fMRI session as Pearson's correlation between each pair of parcels, resulting in a symmetric 268 × 268 matrix. These two FC matrices were further averaged resulting in one FC matrix per subject. One half of the symmetric matrix as well as the diagonal were discarded so that only unique edges were used as features in the prediction workflow.

### Prediction analysis

First, we aimed to reproduce Finn *et al.* [19] prediction pipeline using the CBPM framework and the Leave-One-Out Cross-Validation (LOO-CV) scheme. Specifically, we reconstructed the workflow used to reproduce Figure 5a in Finn *et al.* [19]. As a prediction target, we used subjects' score on the Penn Matrix Test (PMAT24_A_CR). This is a non-verbal reasoning assessment and a measure of fluid intelligence. CBPM first performs correlation-based univariate feature selection based on a pre-specified significance threshold. Selected features are further divided into positively and negatively correlated features and then separately summed up resulting in two features. Subsequently, a linear regression is fitted either on both or one of these features based on user preferences. The results here were obtained using the positive-feature network at a feature selection threshold of $p < 0.01$ in line with Figure 5a from Finn *et al.* [19]. We observed a similar trend in our results albeit with a lower correlation between observed and predicted values (see Figure 7). In addition, we also provide results for a 10-Fold cross-validation with 10 repeats. In this analysis, we also tested CBPM using positive- and negative-feature networks individually as well as both feature networks combined with varying thresholds for feature selection (0.01, 0.05, 0.1).

## DISCUSSION

Julearn aims to bridge the gap between domain expertise in neuroscience and the application of ML pipelines. Toward that goal, julearn provides a simple interface using two key API points only. First, the `run_cross_validation` function provides functionalites to evaluate common ML pipelines. Second the `PipelineCreator` provides means to devise complex ML pipelines that can be then evaluated using `run_cross_validation`. Additional functionalities are also provided to guide and help users to inspect and evaluate the resulting CV scores. In fact, julearn provides a complete workflow for ML that has already been used in several publications [21, 22]. Furthermore, the customizability and open-source nature of julearn will help it grow and extend its functionality.



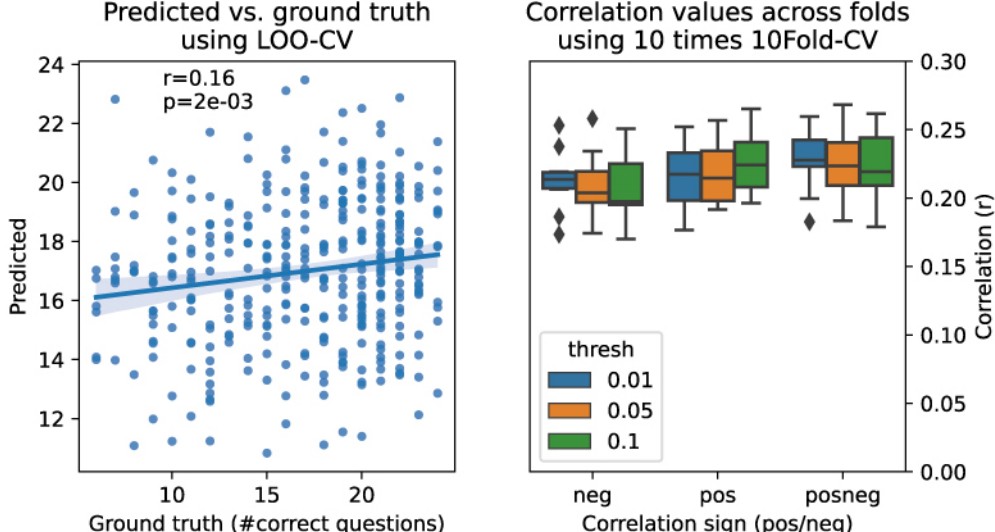

**Figure 7.** Results of the prediction of fluid intelligence using CBPM on HCP-YA data as in Finn *et al.* [19]. Left panel depicts the predicted (*y*-axis) vs the ground truth (*x*-axis) values for each sample in a LOO-CV scheme, following Figure 5a in Finn *et al.* [19]. Right panel depicts the mean correlation values (*r*) across folds, for a 10-times 10-fold CV scheme, using different thresholds (colors) and considering either negative correlations, positive correlations, or both kinds of correlations (columns).

Julearn does not aim to replace core ML libraries such as scikit-learn. Rather, it aims to simplify the entry into the ML world by providing an easy-to-use environment with built-in guards against some of the most common ML pitfalls, such as data leakage that can happen due to not using nested cross-validation and when performing confound removal. Furthermore, julearn is not created to compete with AutoML approaches [23–25], which try to automate the preprocessing and modelling over multiple algorithms and sets of hyperparameters. While these approaches are valid and powerful, they do not offer the full functionalities required in many bio-medical research fields, such as nested cross validation and confound removal. Furthermore, a researcher might require more control over model types, parameters and interpretability, which might not be easily achievable with the current AutoML libraries. Lastly, there are other libraries, such as photon [26], Neurominer [27] or Neuropredict [28], that try to build on top of powerful ML libraires to create different interfaces with unique features for field experts. All these libraries are important for a vibrant open-source community. Hence, julearn's unique features and simple interface will be useful for many research projects.

## AVAILABILITY OF SOURCE CODE

Julearn's code is available in GitHub [29] with the corresponding documentation in GitHub Pages [30]. The code used for the examples in this manuscript is available at [31], with instructions on how to get the publicly available data.

- Project name: Julearn
- Project home page: https://juaml.github.io/julearn/
- Operating system(s): Platform independent
- Programming language: Python
- License: GNU AGPLv3
- RRID: SCR_024881
- biotools: julearn

## DATA AVAILABILITY

The data used in this manuscript is publicly available following each dataset requirements. Information on the dataset sources is provided in the description of each example. Snapshots of the underlying code are available in the GigaDB repository [32].

## LIST OF ABBREVIATIONS

API, Application Programming Interface; CBPM, Connectome Based Predictive Modelling; CV, cross-validation; FC, Functional Connectivity; GM, Gray Matter; GPR, Gaussian Process Regression; HCP-YA, Human Connectome Project Young-Adult; IXI, Information eXtraction from Images; LOO-CV, Leave-One-Out Cross-Validation; MAE, Mean Absolute Error; ML, machine learning; MRI, Magnetic Resonance Imaging; PCA, Principal Component Analysis; rs-fMRI, resting-state functional Magnetic Resonance Imaging; RVR, Relevance Vector Regression; SVR, Support Vector Regression; T1w, T1-weighted.

## DECLARATIONS

### Ethics approval and consent to participate

The authors declare that ethical approval was not required for this type of research.

### Competing interests

The authors declare that they have no competing interests.

### Consent for publication

Consent for publication was obtained from the Alzheimer's Disease Neuroimaging Initiative (ADNI; https://adni.loni.usc.edu/) Data and Publications Committee. Other datasets do not require consent for publication.

### Authors' contributions

SH and FR designed the library. SH, LS, VK, SM, KRP and FR contributed to the development and testing of the library, wrote and reviewed the manuscript. VK contributed to the structural design and writing of julearn's documentation. SM and FR wrote the code for Example 1, SH and FR wrote the code for Example 2 and LS wrote the code for Example 3.

### Funding

This work was partly supported by the Helmholtz-AI project DeGen (ZT-I-PF-5-078), the Helmholtz Portfolio Theme "Supercomputing and Modeling for the Human Brain" the Deutsche Forschungsgemeinschaft (DFG, German Research Foundation), project PA 3634/1-1 and project-ID 431549029–SFB 1451 project B05, the Helmholtz Imaging Platform and eBRAIN Health (HORIZON-INFRA-2021-TECH-01).



Data collection and sharing for this project was funded by the Alzheimer's Disease Neuroimaging Initiative (ADNI) (National Institutes of Health Grant U01 AG024904) and DOD ADNI (Department of Defense award number W81XWH-12-2-0012). ADNI is funded by the National Institute on Aging, the National Institute of Biomedical Imaging and Bioengineering, and through generous contributions from the following: AbbVie, Alzheimer's Association; Alzheimer's Drug Discovery Foundation; Araclon Biotech; BioClinica, Inc.; Biogen; Bristol-Myers Squibb Company; CereSpir, Inc.; Cogstate; Eisai Inc.; Elan Pharmaceuticals, Inc.; Eli Lilly and Company; EuroImmun; F. Hoffmann-La Roche Ltd and its affiliated company Genentech, Inc.; Fujirebio; GE Healthcare; IXICO Ltd; Janssen Alzheimer Immunotherapy Research & Development, LLC; Johnson & Johnson Pharmaceutical Research & Development LLC; Lumosity; Lundbeck; Merck & Co., Inc.; Meso Scale Diagnostics, LLC; NeuroRx Research; Neurotrack Technologies; Novartis Pharmaceuticals Corporation; Pfizer Inc.; Piramal Imaging; Servier; Takeda Pharmaceutical Company; and Transition Therapeutics. The Canadian Institutes of Health Research is providing funds to support ADNI clinical sites in Canada. Private sector contributions are facilitated by the Foundation for the National Institutes of Health (www.fnih.org). The grantee organization is the Northern California Institute for Research and Education, and the study is coordinated by the Alzheimer's Therapeutic Research Institute at the University of Southern California. ADNI data are disseminated by the Laboratory for Neuro Imaging at the University of Southern California.

## Acknowledgements

We want to thank the INM-7 and early adopters of julearn for their valuable contribution at early stages, shaping the direction of our efforts in developing this tool.

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
