## [Editor Report]

Editor’s AssessmentThis Technical Release (Software) paper presents Julearn, an open-source Python library, that allow neuroscience researchers to design and evaluate complex machine learning (ML) pipelines without encountering in common pitfalls such as data leakage and overfitting of hyperparameters. Created to be easy-to-use, accessible for researchers with diverse backgrounds, and to create reproducible results. Bridging the gap between domain expertise in neuroscience and application of ML pipelines. Towards that goal, julearn provides a simple interface only using two key API points. After some debugging and improvements to the documentation testing and review was positive, and a few useful examples are provided in the paper. Additional functionalities are also provided to guide and help users to inspect and evaluate the resulting cross validation scores.

---

## [Reviewer Report]

Reviewer name and names of any other individual's who aided in reviewerJuntao LiuDo you understand and agree to our policy of having open and named reviews, and having your review included with the published manuscript. (If no, please inform the editor that you cannot review this manuscript.)YesIs the language of sufficient quality?YesPlease add additional comments on language quality to clarify if neededIs there a clear statement of need explaining what problems the software is designed to solve and who the target audience is? YesAdditional CommentsIs the source code available, and has an appropriate Open Source Initiative license <a href="https://opensource.org/licenses" target="_blank">(https://opensource.org/licenses)</a> been assigned to the code?YesAdditional CommentsAs Open Source Software are there guidelines on how to contribute, report issues or seek support on the code?YesAdditional CommentsIs the code executable?YesAdditional CommentsIs installation/deployment sufficiently outlined in the paper and documentation, and does it proceed as outlined?YesAdditional CommentsIs the documentation provided clear and user friendly?YesAdditional CommentsIs there enough clear information in the documentation to install, run and test this tool, including information on where to seek help if required?YesAdditional CommentsIs there a clearly-stated list of dependencies, and is the core functionality of the software documented to a satisfactory level?YesAdditional CommentsHave any claims of performance been sufficiently tested and compared to other commonly-used packages? YesAdditional CommentsIs test data available, either included with the submission or openly available via cited third party sources (e.g. accession numbers, data DOIs)?YesAdditional CommentsAre there (ideally real world) examples demonstrating use of the software? YesAdditional CommentsIs automated testing used or are there manual steps described so that the functionality of the software can be verified?YesAdditional CommentsAny Additional Overall Comments to the AuthorJulearn is an open-source Python library that allows domain experts without in-depth ML and programming language training to design and evaluate complex ML pipelines without encountering in common pitfalls, such as data leakage and overfitting of hyperparameters. I acknowledged it is a user-friendly software and deserves more attention. Yet, I have some minor concerns.  1. It is advisable to include the documentation website of the software in abstract, as comprehensive documentation serves as a crucial resource for users seeking to utilize the software effectively and efficiently.  2. When I run the first example mentioned in the manuscript, I got some problems below: 1) I run the first step, 1_get_data.py to download the data to the path of data/1_brain_age. But when I executed the second step, 2_predict_brain_age.py, I got the error “FileNotFoundError: [Errno 2] No such file or directory: 'data/ixi.S4_R8.csv'”. I changed the code in line 19 of the 2_predict_brain_age.py, data_dir=Path(__file__).parent.parent/"data", to data_dir=Path(__file__).parent.parent/"data"/"1_brain_age". Then it can run successfully. 2) Before I run the 2_predict_brain_age.py, I got the “ModuleNotFoundError: No module named 'skrvm'” error. I don’t know how to install this python library until check in the 2_predict_brain_age.py file. Providing a README file within the directory of each example that details the steps of executing the example code would be useful, as it would allow users to easily download the data required and comprehend the installation process for required libraries and dependencies. 3) As well as the second example, I need to check the 1_prcess_data.py to known how to download the ADNI data. Still you can write a readme file for each example.  3. I found that Figure 3 appears before Figure 2. Maybe you can change the orders of this two Figures.  4. The codes in Figure 1 and 2 are intended to show the differences between julearn and sklearn. Instead of the Figures, you can just put the code in an in-line text box, maybe it is clearer. Furthermore, code annotations can be included to provide guidance on potential applications of this code to users.  5. In the Inspection and Analysis section, you mentioned that Julearn includes two functionalities: a preprocess_until function and Inspector class. Maybe you can also provide a code in an in-line text box to demonstrate how these two functionalities can be used.RecommendationMinor Revisions

---

## [Reviewer Report]

Reviewer name and names of any other individual's who aided in reviewerShuangsang FangDo you understand and agree to our policy of having open and named reviews, and having your review included with the published manuscript. (If no, please inform the editor that you cannot review this manuscript.)YesIs the language of sufficient quality?YesPlease add additional comments on language quality to clarify if neededIs there a clear statement of need explaining what problems the software is designed to solve and who the target audience is? YesAdditional CommentsIs the source code available, and has an appropriate Open Source Initiative license <a href="https://opensource.org/licenses" target="_blank">(https://opensource.org/licenses)</a> been assigned to the code?YesAdditional CommentsAs Open Source Software are there guidelines on how to contribute, report issues or seek support on the code?YesAdditional CommentsIs the code executable?Unable to testAdditional CommentsIs installation/deployment sufficiently outlined in the paper and documentation, and does it proceed as outlined?Unable to testAdditional CommentsIs the documentation provided clear and user friendly?YesAdditional CommentsAdditional CommentsIs there a clearly-stated list of dependencies, and is the core functionality of the software documented to a satisfactory level?YesAdditional CommentsHave any claims of performance been sufficiently tested and compared to other commonly-used packages? YesAdditional CommentsIs test data available, either included with the submission or openly available via cited third party sources (e.g. accession numbers, data DOIs)?YesAdditional CommentsAre there (ideally real world) examples demonstrating use of the software? YesAdditional CommentsAdditional CommentsAny Additional Overall Comments to the AuthorThe paper titled "Julearn: an easy-to-use library for leakage-free evaluation and inspection of ML models" by Sami Hamdan et al. introduces a library that empowers researchers to design and evaluate complex ML pipelines. This library provides users with a user-friendly environment that incorporates safeguards against common ML pitfalls. Consequently, it offers convenience and usefulness to its users. Nonetheless, there are a few concerns that need to be addressed: 1.The authors should clearly articulate the relationship between Julearn and Scikit-learn (sklearn) and perform a comparative analysis of their shared and distinctive features. 2.It would be beneficial to include a table or figure that provides a comprehensive list of functions or ML models available in Julearn, enabling users to quickly familiarize themselves with the library's capabilities. 3.While the Visualization component of Julearn currently only offers the "plot_scores" function, the inclusion of additional plotting functions would be advantageous in providing users with a more comprehensive visualization toolkit.  By addressing these concerns, the usability and effectiveness of Julearn can be further enhanced, ensuring a more robust and user-friendly experience for researchers utilizing the library.RecommendationMinor Revisions